# Implementation of National Nutrition Policies and Strategies to Reduce Unhealthy Diets: An Ecological Analysis of 194 Countries from 2017 to 2021

**DOI:** 10.3390/nu16060911

**Published:** 2024-03-21

**Authors:** Alina Ioana Forray, Cristina Maria Borzan

**Affiliations:** Discipline of Public Health and Management, Department of Community Medicine, Iuliu Hațieganu University of Medicine and Pharmacy, Victor Babeș 8, 400347 Cluj-Napoca, Romania

**Keywords:** nutrition policies, unhealthy diets, non-communicable diseases, policy implementation, dietary strategies, public health nutrition, health policy, World Health Organization

## Abstract

This study investigates the implementation of national policies and strategies to control unhealthy diets, which are pivotal in the global surge of non-communicable diseases. Leveraging data from the World Health Organization’s Non-Communicable Diseases Progress Monitors and Country Capacity Surveys, we calculated aggregate implementation scores for 13 diet-related policies across 194 countries from 2017 to 2021. We used descriptive statistics and linear regression to investigate the implementation trends and associations between key national-level factors and implementation scores. The mean score in 2021 was 52% (SD = 24), with no statistical differences in the 5-year period. Stark disparities in implementation efficacy were noted, ranging from comprehensive adoption in some nations to minimal application in others. Our analysis also highlights a shifting focus in policy adoption: notably, an increased commitment to taxing sugar-sweetened beverages juxtaposed with a decline in dietary awareness initiatives. Significant predictors of policy implementation include the Human Development Index, the cost of a healthy diet, and health service coverage. These findings suggest a complex interdependence of socioeconomic factors influencing policy implementation. Our research underscores the need for multifaceted, globally collaborative strategies to effectively combat diet-related diseases, emphasizing the importance of comprehensive policy frameworks in public health interventions.

## 1. Introduction

Findings from the 2019 Global Burden of Disease Study [1], analyzing data from 204 countries between 1990 and 2019, conclude that over the past 35 years, the top three causes of death globally are non-communicable diseases (NCDs) that are related to modifiable lifestyle factors, such as unhealthy diet. NCDs, responsible for 74% of global deaths annually, are significantly influenced by modifiable behavioral risk factors such as tobacco use, excessive sodium intake, alcohol consumption, and a lack of physical activity, alongside metabolic risk factors like hypertension, obesity, hyperglycemia, and hyperlipidemia [2]. The global burden of NCDs is not limited to high-income countries, as over 86% of premature deaths occur in low-and middle-income countries, presenting a substantial public health challenge worldwide [2].

National nutrition policies play a crucial role in combating unhealthy diets and NCDs by targeting dietary behaviors and promoting healthy diets [3]. The World Health Organization (WHO) has developed strategies to improve diets and overall health [4]. A global nutrition policy review found that 77% of surveyed member states had comprehensive nutrition policies to address different forms of malnutrition. However, implementing globally recommended policies for addressing non-communicable diseases caused by unhealthy diets has been challenging. Policy advocates face strong opposition from the food industry, pressure to minimize trade impacts, and a lack of data to demonstrate the link between policies and health outcomes [5]. The effectiveness of these policies varies across countries, with some lacking specificity, budgetary commitments, and accountability mechanisms [6]. Most of the time, existing policies in high-income countries often focus on individual responsibility and lack comprehensive, inter-sectoral approaches [7]. The influence of food environments on dietary choices underscores the need for comprehensive, multilevel food policies [8]. To combat diet-related NCDs, policymakers need to prioritize public well-being over commercial interests. They should adopt a comprehensive perspective that incorporates a systems approach, health equity, and global collaboration [9].

A thorough analysis of the implementation status of nutrition-related policies and strategies and national factors can help us understand how nutrition policies are implemented. Identifying the factors that influence policymaking and implementation can improve our understanding of how nutrition policies are formulated within larger political and economic contexts. Lastly, it can identify opportunities and constraints for integrated nutrition policy. 

The WHO regularly evaluates a country’s ability to prevent and control NCDs using a global survey called the NCD Country Capacity Survey (CCS), which has been in effect since 2001. This assessment helps countries and the WHO to track progress and accomplishments in strengthening their capacities to tackle the growing issue of NCDs [10,11,12]. The WHO issues NCD Progress Monitor reports every two years to evaluate the compliance of global policies with the “best buys” policies adopted in 194 countries. They include country profiles and an evaluation of the full, partial, or non-implementation of each policy in every country [13,14,15]. These reports classify policies into four time-bound commitments, which were adopted during the UN High-Level Meetings [16]. These reports and surveys on NCDs are filled by national NCD focal points or designated colleagues within the Ministry of Health or a national institute/agency. The NCD CCS questionnaire covers different topics such as health system infrastructure, funding, policies, plans and strategies, surveillance, primary health care, partnerships, and multilateral collaboration. Several questionnaire components represent measures addressing an unhealthy diet [17]. 

Despite the availability of progress monitors and global surveys, there seems to be a lack of detailed analysis on the worldwide implementation of nutrition-related policies, as well as the key factors that predict successful implementation. To the best of our knowledge, these data have not been studied to understand the patterns in implementing national nutrition policies. However, the WHO progress monitors and reports of the NCD CCS are a significant source of insight. Building on previous research on NCD policy implementation across countries [18,19], our study aims to fill two specific gaps: the focus on nutrition-focused policies, including policies available from the WHO Country Capacity Surveys, and extending the analysis period from 2017 to 2021, offering a comprehensive view of global health policy progress.

In this ecological analysis encompassing 194 countries, our study aimed to dissect the global landscape of national nutrition policies and strategies and their implementation in mitigating unhealthy diets from 2017 to 2021. We sought to quantify the range and mean implementation of diet-related NCD policies globally, identify the most universally adopted strategies, and highlight the disparities in policy implementation across nations. Our investigation extended to discerning trends over time, examining the dynamics of policy adoption and discontinuation, and elucidating the impact of socioeconomic, health status, and risk factors on policy implementation. Through this nuanced analysis, we endeavored to provide a critical evaluation of the current state of nutrition policy implementation, offering insights into the complex interplay of factors that drive or hinder progress in the global fight against diet-related health issues. This research contributes to the foundational understanding necessary for crafting more targeted, effective public health strategies and policies aimed at reducing the burden of diet-related NCDs worldwide.

## 2. Materials and Methods

### 2.1. Study Design

In this study, we adopted an ecological study design to examine the implementation of nutrition policies and strategies across 194 countries, focusing on their effectiveness in reducing unhealthy diets. By utilizing data from the WHO’s NCD Progress Monitors and NCD CCS, we gained insights into national efforts and their outcomes in addressing dietary health challenges. This approach allows for the analysis of population-level data, facilitating a comprehensive understanding of how different nations navigate the complexities of public health nutrition policy and strategy implementation. 

### 2.2. Data and Measurement 

For this study, we extracted data collected by the WHO to monitor the progress of countries in implementing NCD policies in its 194 member states, specifically the results from the NCD CCS and NCD Progress Monitors in 2017, 2019, and 2021. The NCD CCS is designed to evaluate a country’s capacity to prevent and manage NCDs through a comprehensive and structured data collection instrument. The surveys were meticulously designed to encompass a broad range of data across four domains: the establishment of public health infrastructure; the formation of partnerships and engagement in multisectoral collaboration; the development and execution of policies, strategies, and action plans; the fortification of health information systems; and the monitoring, surveillance, and health system capacity for the detection, treatment, and care of NCDs. Detailed methodologies pertaining to the development of the questionnaire, the nuances of the data collection process, and the rigorous data validation protocols are thoroughly documented in the annual reports issued by the WHO [10,11,12,13,14,15].

The indicators derived from the NCD Progress Monitors were quantified using a scoring system, where a fully achieved indicator was allocated a full point, a partially achieved indicator received a half point, and a non-achieved indicator was assigned zero points. Additionally, responses from the NCD CCS that pertained exclusively to the capacity surveys were scored with a binary system, where a positive response was given one point, and a negative response received no points. 

Our approach involves calculating national aggregate scores for the years 2017, 2019, and 2021, which are based on thirteen indicators reflecting various dimensions of diet-related NCD prevention and control. We summed the total implementation scores for all diet-related policies to create an aggregate score for policy implementation (ASPI). To facilitate comparative analysis across countries, we normalized the ASPI values against a theoretical maximum of 13 points, which corresponds to the policy indicators being investigated, converting the results into percentages.

In our study, we selected thirteen components from the WHO survey, focusing on those that directly impact the implementation and efficacy of national nutrition policies aimed at reducing unhealthy diets (Table 1). This curated inclusion, encompassing indicators such as national targets, data collection systems, and specific nutrition policies like sodium reduction and trans fat elimination, was designed to closely align with our research objectives centered on diet-related non-communicable diseases (NCDs). Conversely, we excluded broader NCD prevention measures, including tobacco and alcohol controls and physical activity enhancements, to maintain a focused analysis of dietary policies. The complete list of indicators included in the WHO’s NCD Progress Reports and CCS is available in the Appendix A.

### 2.3. Statistical Analysis

The analysis utilized descriptive statistics to explore the extent of policy implementation for each diet-related policy and the ASPI among the 194 countries in 2021. An in-depth review of the aggregate scores highlighted countries with the highest and lowest implementation scores in each year and the differences in the five-year period. To visually summarize these distributions, boxplots were generated to display the mean implementation scores segmented by World Bank income groups and geographical regions.

To assess the dynamic changes in policy implementation over time, we categorized countries according to the changes in their ASPI from 2017 to 2021. This categorization allowed for the observation of significant trends in policy adoption over the five-year period. Heat maps facilitated a comprehensive presentation of the implementation scores for all countries, and clustered bar charts depicted the scores across the thirteen policy indicators.

The study’s analysis was underpinned by a theoretical framework that considers socioeconomic, health-related, and dietary risk factors as explanatory variables due to their relevance to NCD policy outcomes (Table 2) [19]. Linear regression analyses were conducted to investigate the associations between the policy implementation scores and these variables, using both univariate and multivariate approaches. This analysis aimed to quantify the impact of each explanatory variable on policy implementation while controlling for other factors in the model. The regression models treated explanatory variables as continuous, and the normality of the residuals was confirmed, ensuring the appropriateness of linear regression techniques for our data. 

Our examination of the implementation of national nutrition policies revealed disparities in data availability across 13 specified policy areas. Notably, the gaps ranged from 0.2% for conducting a STEPS survey or a comprehensive health examination survey every five years to 3.5% for the existence of a tax on sugar-sweetened beverages. Three policies were available only for 2021 (tax on foods with high sugar/fat/salt content, price subsidies for healthy food, and surveillance of dietary risk factors through a national survey). These were included in the descriptive analysis for the implementation situation in the year 2021 and excluded from the aggregate score to ensure comparability in the temporal analysis.

Statistical significance was established at a *p*-value threshold of less than 0.05. IBM SPSS Statistics for Mac OS (Version 29.0.1) and RStudio Desktop for MacOS (Version 2023.06 “Mountain Hydrangea”) served as the primary tools for statistical computation. For data visualization, the ggplot2 package was utilized, ensuring that the graphical representations of the data were both informative and accessible. This study did not require ethical approval, as all data are publicly available. 

## 3. Results

In our examination of the global implementation of national nutrition policies over the period from 2017 to 2021, encompassing 194 countries, we documented a gradual increase in policy implementation. The data for 2017 revealed an average ASPI of 49.8% (SD = 23.78), with the range of scores extending from 0% in several low-income countries to a full score of 100% in Brazil. By 2019, there was a slight elevation in the average score to 49.94% (SD = 24.86), with the highest scores recorded at 96.15% in Norway and Finland. The trend continued into 2021, as the average ASPI further improved to 51.98% (SD = 23.94), with scores ranging from 3.85% in six low-income and lower-middle income countries to 96.15% in Turkey, the UAE, and Finland. Despite these variations, the analysis conducted through a one-way ANOVA showed no significant difference in the aggregate mean scores over the years, highlighting a consistent global average in the effectiveness of policy implementation. The distribution of scores in each year was normally distributed with a small right skew. 

In the analysis of national nutrition policy implementation spanning from 2017 to 2021, notable trends emerge regarding the prevalence and adoption of specific policies (Figure 1, Appendix A). The most implemented policy by 2021 was the “Comprehensive Diet and Nutrition Plan”, achieving an implementation rate of 83.5% (*n* = 162). This policy consistently showed the highest implementation rates across the observed period, highlighting its central role in global nutrition and public health strategies. Conversely, the least widely implemented policy by 2021 was “Price Subsidies for Healthy Foods”, with an implementation rate of 7.7% (*n* = 15). This indicates significant challenges and global reluctance to adopt fiscal measures aimed at promoting healthier dietary choices through subsidies. The analysis also revealed significant shifts in policy focus, as evidenced by the “Dietary Awareness Efforts” policy, which experienced a notable reduction in full implementation from 77.3% (*n* = 150) in 2017 to 55.7% (*n* = 108) in 2021. In contrast, the “Sugar-sweetened Beverages Tax” saw an increase in full implementation from 23.2% (*n* = 45) in 2017 to 46.9% (*n* = 91) in 2021, reflecting a growing global commitment to implementing fiscal policies targeting unhealthy diets. Full results for each policy and country by year are available in the Appendix A.

The data for 2021 indicate a considerable variation in the adoption of dietary policies aimed at reducing non-communicable diseases across the 194 countries examined. Notably, the United Arab Emirates, Turkey, and Finland each achieved an implementation score of 96.15% (12.5 points out of a total of 13). In contrast, countries at the lower end of the spectrum, such as Yemen, Somalia, and Libya, recorded implementation scores as low as 3.85% (0.5 points). Ukraine exhibited the most substantial progress, with its implementation score surging from 11.54% to 76.92%, indicating a vigorous enhancement in its dietary policy framework. Similarly, the Philippines and Kiribati each saw an increase in their scores of 53.85%, signaling strong policy improvements. Nigeria, Tuvalu, and Senegal also demonstrated significant strides, with each country’s score rising by 42.31%. Meanwhile, Saint Vincent and the Grenadines recorded a 38.56% increase, marking its commitment to dietary policy development (Figure 2, Appendix A).

The analysis reveals a dynamic landscape of policy implementation characterized by gains and regressions in specific policy areas and countries. Notably, the data indicate that 49.0% of countries experienced an increase in their ASPI, 37.1% witnessed a regression, and 13.9% saw no change (Appendix A).

From 2017 to 2021, our analysis of global nutrition and NCD policy implementations highlights significant shifts, both positive and negative. At the forefront of progress, the implementation of taxes on sugar-sweetened beverages (SSBs) exhibited the greatest increase, with a remarkable upswing of 22.35%. This was followed by the establishment of national multisectoral commissions for NCDs, which saw an 11.89% rise, and the operationalization of NCD units within health ministries, with a 9.84% improvement. Additionally, the adoption of operational diet plans and the setting of WHO-guided national targets improved by 4.76% and 5.32%, respectively.

Conversely, the landscape also revealed regressions, particularly in public health communication, where dietary improvement awareness campaigns decreased by −21.43%. Notably, dietary awareness campaigns emerged as the area with the most pronounced regression, affecting 47 countries, an indicator of shifting priorities or challenges in public health messaging. Efforts towards reducing population salt consumption and addressing saturated/trans fatty acids in diets also saw declines of −3.53% and −1.09%, respectively. Salt reduction and fat content policies encountered regressions in 38 and 28 countries, respectively, signaling potential areas for reinvigorated focus and policy innovation. Moreover, slight regressions were noted in the marketing of foods to children and the implementation of the International Code of Marketing of Breast Milk Substitutes, with changes of 2.72% and −4.35% (Figure 3).

ASPI progressively increased from low-income (mean = 29.3%, SD = 14.9) to high-income countries (mean = 64.8, SD = 20.5), highlighting a robust association between a nation’s economic capacity and its implementation of nutrition policies (Figure 4). This disparity in ASPI was statistically significant, as indicated by the ANOVA results (F(3, 190) = 19.129, *p* < 0.001), with approximately 23% of the variance of ASPI explained by differences between groups, as denoted by Eta-squared (0.232).

Furthermore, a regional analysis revealed significant disparities in ASPI among geopolitical regions, with the lowest scores observed in Sub-Saharan Africa (mean = 32.93, SD = 18.26) and the highest in North America (mean = 75.0, SD = 2,71) (F(6, 187) = 11.942, *p* < 0.001). The effect sizes for this analysis (Eta-squared = 0.277) underscore the substantial impact of regional factors on the effectiveness of nutrition policies (Figure 5). 

In our univariate analysis of determinants influencing the aggregate scores for national policy implementation across 194 countries, variables were categorized into educational, economic, NCD burden, and dietary risk factors to delineate their impact on the implementation of nutrition policies (Table 3). The Human Development Index (HDI) was identified as the most substantial positive predictor of the aggregate score, showing a significant association (R^2^ = 0.382, *p* < 0.001), indicating the essential role of broad human development facets in the implementation of health policies. The Universal Health Coverage (UHC) Index also demonstrated a significant positive relationship with the implementation score (R^2^ = 0.375, *p* < 0.001). This indicates the importance of broad health service coverage in supporting the implementation of comprehensive diet policies. Conversely, economic accessibility, as evidenced by the cost of a healthy diet (R^2^ = 0.029, *p* < 0.001) and the inability to afford a healthy diet (R^2^ = 0.401, *p* < 0.001) negatively influenced implementation scores, underscoring the importance of affordability in policy implementation.

NCD burden-related variables such as the NCD mortality rate and premature mortality from NCDs showed significant associations with implementation scores, with the NCD mortality rate presenting a negative effect (R^2^ = 0.124, *p* < 0.001) and premature mortality from NCDs indicating potential challenges in leveraging nutrition policies to combat NCD mortality effectively (R^2^ = 0.125, *p* < 0.001).

Dietary intake factors like alcohol intake and saturated fat intake were positively associated with implementation scores (alcohol intake: R^2^ = 0.082, *p* < 0.001; saturated fat: R^2^ = 0.015, *p* = 0.004), highlighting the nuanced relationship between dietary habits and policy implementation. Sodium and added sugars intake further contributed to this complex picture, with sodium intake showing a moderate positive correlation with implementation scores (R^2^ = 0.119, *p* < 0.001) and added sugars intake exhibiting a weaker positive effect (R^2^ = 0.009, *p* = 0.024), emphasizing the recognition of specific nutrients in national nutrition policy frameworks. 

These findings, derived from a rigorous and structured examination of individual contributions to the aggregate policy scores, underscore the multifaceted nature of influences on diet policy implementation. The subsequent multivariate regression analysis confirmed the enduring significance of the HDI, the cost and affordability of a healthy diet, premature mortality from NCDs, and the UHC Service Coverage Index as pivotal determinants, with the model accounting for a considerable portion of the variance in the aggregate scores (R^2^ = 0.550, *p* < 0.001) (Figure 6). This analysis not only reaffirms the critical roles of human development, economic access, and health service coverage, but also highlights the complex interplay between socioeconomic, health, and educational factors in shaping effective national nutrition policies.

## 4. Discussion

In the evolving landscape of global public health, the implementation of national nutrition policies stands as a critical frontier in the battle against NCDs, which are predominantly fueled by unhealthy diets and lifestyle choices [3]. Our comprehensive analysis, spanning 194 countries from 2017 to 2021, unveils a nuanced picture of the global efforts to combat diet-related health challenges. Despite the overarching presence of dietary risk factors, which have caused a significant increase in all-cause deaths and disability-adjusted life years (DALYs) worldwide since 1990 [34], our findings highlight a gradual yet inconsistent improvement in policy implementation effectiveness across nations. The increase in average ASPI from 49.80% in 2017 to 51.98% in 2021, while modest, indicates a slow but positive shift towards more robust nutrition policy frameworks worldwide. This increment, which is statistically non-significant, underscores the possible constraints of national health governance structures to enhance dietary health policies amidst a plethora of challenges, including strong opposition from the food industry, leadership deficits, insufficient political will, resource scarcity, inadequate monitoring, and poor accountability [35,36,37].

In this study, we introduce the methodology for an aggregate score (ASPI) to evaluate the implementation of national nutrition policies aimed explicitly at mitigating the impact of unhealthy diets. This endeavor is distinct from previous frameworks, notably diverging from the WHO’s Nutrition Governance Score [38], which, while comprehensive, has not been updated since 2018. Furthermore, it refines the scope beyond the broader lens of NCD policy evaluation found in Allen et al.’s work [18,19] by focusing exclusively on dietary strategies. Utilizing the WHO’s robust data, ASPI offers a focused, evidence-backed insight into the effectiveness of diet-related policy implementation, filling a crucial gap in the literature. Through this refined lens, our study contributes to the critical discourse on enhancing nutrition policy frameworks.

Examining specific policy implementations reveals a striking disparity in global priorities and actions. The widespread adoption of a national strategic plan, strategy, or operational policy for unhealthy diets, with an 83.51% implementation rate by 2021, signifies a global consensus on the importance of holistic approaches to nutrition and public health. This policy’s prominence underscores that most countries globally have acknowledged the importance of combating unhealthy diets. By investigating the national diet policies in the WHO NCD Document Repository [39], we can recognize that these policies more often focus on promoting individual healthy choices, and there is a lack of comprehensive, multi-strategy policies to address the complex web of factors that influence the rise in diet-related NCDs [7]. Other policies that were most widely implemented are risk factor surveys, the existence of control units, targets, and operational NCD plans. In contrast, policies regarding fat, salt, and sugar reduction policies and food marketing were the least adopted policies. One possible reason for the difference between plans and policies targeting risk factors is that paper-based policies do not threaten powerful vested interests [19].

In stark contrast, in 2021, the implementation of price subsidies for healthy foods or taxes on food high in fat, sugars, or salt was markedly low, with a mere 7.73%, respectively, a 12.9% implementation rate. This gap illustrates the global reluctance to engage in fiscal interventions that directly influence consumer behavior toward healthier dietary choices due to the cross-sectoral nature of the interventions, with health and economic policymakers having different priorities and evidence considerations [40]. Despite these challenges, there is evidence that fiscal interventions can influence dietary behaviors, with a systematic review suggesting that a minimum of 10–15% tax on unhealthy foods and subsidies for healthy ones can be effective [41]. However, an important shift in utilizing fiscal interventions was observed in the implementation of taxes on sugar-sweetened beverages, which increased from 23.20% in 2017 to 46.91% in 2021. These findings not only underscore the evolving landscape of nutrition policy, but also highlight the growing evidence on the effect of various types of SSB taxes [42] and significant public support for government action [43], although there still exists challenges to implementation due to industry opposition and resource constraints [44]. 

The dynamic shifts in policy focus and implementation effectiveness across different countries are indicative of the complex interplay between global recommendations and national priorities. The notable progress observed in countries such as Ukraine and the Philippines, with substantial increases in their implementation scores, reflects an encouraging trend of policy adaptation in response to the pressing burden of NCDs. These notable advancements in outlying overperformers most often signal a commitment to improving national dietary frameworks, potentially catalyzed by external threats, such as the war with Russia, which impacted the need for food security, the food supply and consumption in Ukraine, and the need for strategic planning [45], leading to the implementation of te WHO’s policy in Ukraine [46]. In the Philippines, malnutrition represents a complex problem with a high burden of child undernutrition, increasing the pressure on the government to implement nutrition policies [47,48]. 

Conversely, the regression in implementation scores in more than one-third of nations highlights a concerning trend of deprioritizing public health nutrition. This regression could be attributed to operational and administrative incapacity, budgetary constraints, cultural barriers, or political and economic instability [49]. Such trends underscore the need for improving the national capacity of health policies through embedding research into policy and practice and the need for institutional capacity building [50,51]. 

The analysis further sheds light on the stark disparities in ASPI across income groups and regions, revealing a pronounced gradient from low- to high-income countries. This gradient not only illustrates the significant impact of economic capacity on the ability to implement effective nutrition policies, but also highlights the broader socioeconomic challenges facing lower-income nations. The association between a nation’s economic status and its policy implementation effectiveness underscores the critical role of financial resources, infrastructure, and institutional capacity in driving public health initiatives. Moreover, the regional disparities, particularly the lower scores observed in Sub-Saharan Africa compared to higher scores in North America, emphasize the influence of regional geopolitical, economic, and social factors on public health policy implementation. These findings call for a nuanced understanding of the barriers to policy implementation in different contexts and underscore the importance of tailored, context-specific strategies to enhance the global response to diet-related NCDs. Addressing the global challenge of unhealthy diets and diet-related NCDs requires not only increased financial investment, but also the strengthening of public health institutions, the fostering of global collaboration, and the adoption of context-specific strategies that are sensitive to the unique socioeconomic and geopolitical realities of each country [52,53,54].

Our univariate analysis of determinants influencing national policy implementation aggregate scores presents a compelling narrative on the multifaceted influences on nutrition policy implementation. Our multivariate model explained 55% of the variability in ASPI, with four variables retaining their significant associations. Our results indicate that countries with high HDI, high UHC Service Coverage Index, high premature mortality from NCDs and overall mortality from NCDs, and low cost of a healthy diet implement more diet-related policies after adjusting for other financial factors, risk factors, and demographic structures. This points to the importance of overall development and health system performance in supporting effective diet policy frameworks. The HDI emerged as a significant predictor, reinforcing the concept that broader human development facets are integral to the successful implementation of health policies. This relationship underscores the interconnectivity between education, economic prosperity, and health outcomes, highlighting the necessity of a holistic approach to public health policy that transcends traditional health sector boundaries. Conversely, the negative influence of the cost and affordability of a healthy diet on ASPI reveals a critical barrier to policy implementation. This points to the essential role of economic access and affordability in dietary choices and health policy. These findings underscore the importance of addressing economic barriers and ensuring the affordability of healthy diets as a cornerstone of effective nutrition policy.

### Limitations

Our investigation aims to identify national-level correlates within the implementation of national nutrition policies and strategies across a comprehensive dataset spanning 194 countries. However, the current study encounters inherent limitations, primarily in its methodology for aggregating scores for policy implementation. The adopted scoring system, which categorizes the achievement of indicators in a simplified manner—full points for complete achievements, half points for partial ones, and zero for non-achievements—along with a binary assessment for certain responses, may inadvertently reduce the granularity of policy effectiveness evaluation. Such a methodology, despite its alignment with the WHO-established frameworks, risks oversimplifying the diverse and complex nature of policy implementation. This approach treats policies of varying efficacy as equivalent, which could obscure the nuanced effectiveness or comprehensiveness of individual policies. This limitation underscores the broader challenge of quantitatively assessing policy effectiveness through aggregated measures, which may not accurately reflect the intricate contexts, strengths, and weaknesses inherent in the policy frameworks of different nations. 

Furthermore, the study’s reliance on publicly available data sources, specifically the WHO’s NCD Progress Monitors and CCS, introduces a potential for bias stemming from self-reported data. The variability in the accuracy and reliability of such data across countries—affected by disparate data collection methodologies, interpretations of survey questions, and reporting thoroughness—poses a significant challenge. Although the WHO employs rigorous data validation methods, the variability in the enforcement and practical implementation of policies across different geopolitical contexts may lead to a disconnect between the legal frameworks and their practical application. This gap emphasizes a critical limitation in our understanding of the real-world effectiveness of nutrition policies in mitigating NCD risk factors at the population level. Consequently, the study’s findings may project an overly optimistic portrayal of global policy implementation, potentially overlooking the substantive challenges and barriers that impede the enforcement and practical application of these policies. This discrepancy highlights the importance of adopting a more nuanced perspective in evaluating the success of nutrition policies and strategies beyond aggregated self-reported data.

Lastly, the methodological decision to exclude certain countries due to incomplete data or to omit recently introduced policies for the sake of standardizing comparisons introduces an additional layer of complexity in interpreting our findings. This approach, while aimed at enhancing comparability and facilitating temporal analysis, might inadvertently skew the results. By neglecting newer policies and disregarding countries with insufficient data, our analysis may not fully capture the extent of global efforts toward implementing nutrition-related policies. This limitation not only impacts the study’s comprehensiveness, but also its ability to generalize conclusions across different geopolitical contexts. Moreover, the ecological design of the study, focused on national-level data, inherently limits the exploration of the intricate interplay between individual, community, and systemic factors that influence dietary behaviors and policy outcomes. Despite providing valuable insights into the global landscape of policy implementation, this design may overlook significant local and regional nuances that critically affect the success of nutrition policies. Such oversight underscores the necessity for complementary qualitative and mixed-methods research to delve into the country-specific circumstances and factors that shape the effectiveness of the nutrition policy environment and its outcomes. 

Additionally, the analysis’s reliance on a set of predetermined indicators for regression models, while methodologically sound, may not capture the full spectrum of factors influencing policy success, highlighting the inherent trade-offs in global health research between data availability, comparability, and depth of analysis. The reliance on broad indicators such as the UHC, HDI, and literacy rates, though informative, does not capture the nuanced realities of countries’ development. The Global Dietary Database and the assessment of the cost of a healthy diet, constrained by the representativeness of surveys and the availability of detailed food price data, respectively, underscore the complexities of using national-level predictors. These limitations highlight the need for cautious result interpretation and further research to achieve a more comprehensive understanding of the multifaceted factors influencing nutrition policy implementation and effectiveness globally.

## 5. Conclusions

Our study’s examination of national nutrition policies in 194 countries from 2017 to 2021 reveals a positive, albeit modest, trend towards more effective strategies against diet-related NCDs. While we note gradual improvements in policy frameworks and strategy adoption, significant disparities and challenges persist, particularly across different income levels and regions. These findings underscore the necessity for a context-specific, multidisciplinary approach to nutrition policy that encompasses public health, economic, and social considerations. Moving forward, the focus must be on developing integrated policies that tackle the root causes of unhealthy diets and NCDs, emphasizing public well-being, fostering global collaboration, and leveraging evidence-based strategies to reduce the global burden of diet-related NCDs.

## Figures and Tables

**Figure 1 nutrients-16-00911-f001:**
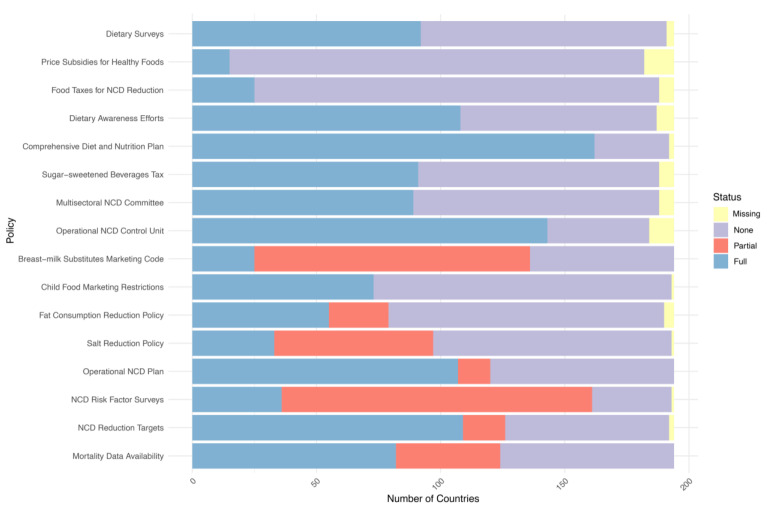
Implementation status of nutrition-related policies and strategies in 2021 (note: NCD, non-communicable disease).

**Figure 2 nutrients-16-00911-f002:**
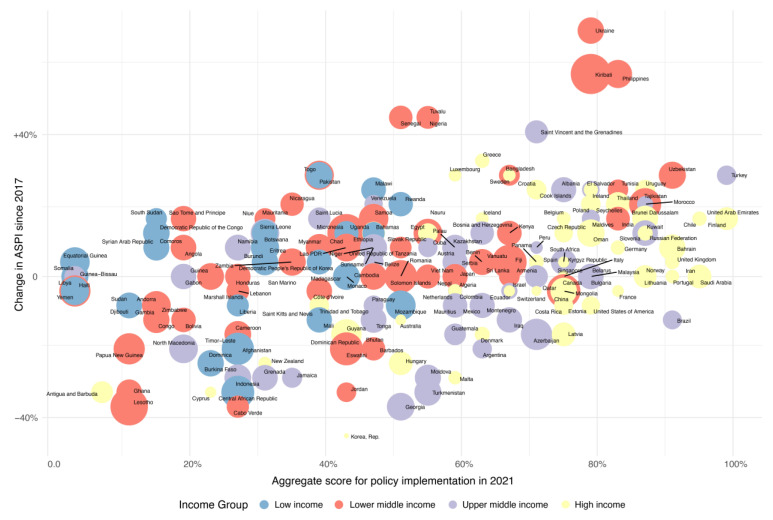
The global aggregate score for policy implementation in 2021 in 183 countries, differentiated by World Bank Income Group, mapped against the percentage change in the score since 2017 (note: Bubble size indicates the NCD mortality rate; NCD, non-communicable disease; ASPI, aggregate score for policy implementation).

**Figure 3 nutrients-16-00911-f003:**
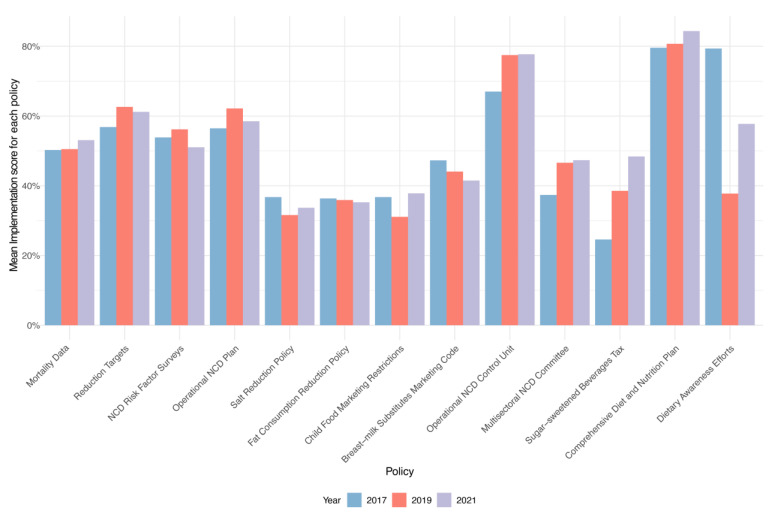
Average implementation of the thirteen nutrition-related policies and strategies by year across 194 countries. (Note: A score of 0 refers to no implementation or no available data; a score of 100% means full implementation across all countries; NCS, non-communicable diseases).

**Figure 4 nutrients-16-00911-f004:**
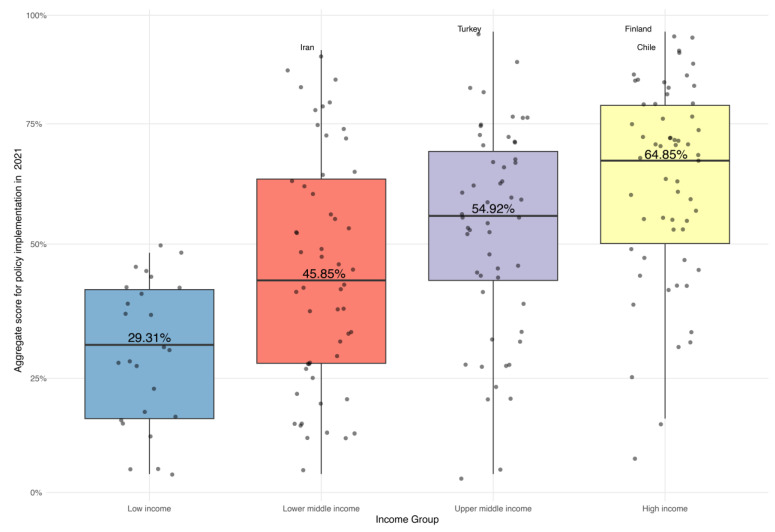
Mean aggregate score for policy implementation by World Bank Income Group in 2021.

**Figure 5 nutrients-16-00911-f005:**
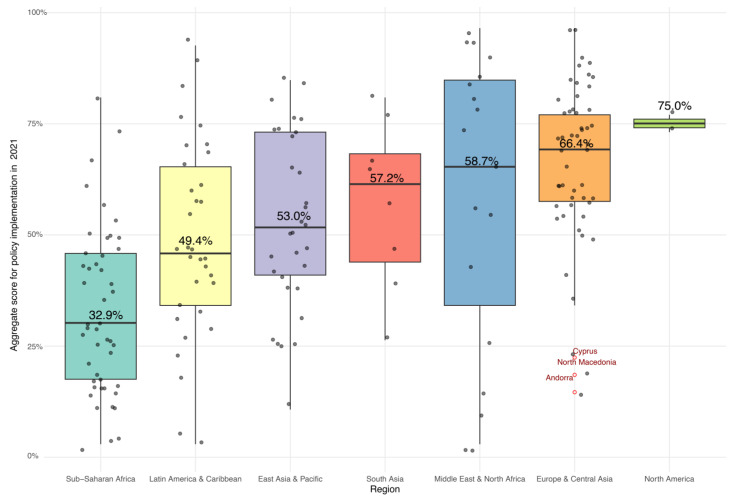
Mean aggregate score for policy implementation by region in 2021.

**Figure 6 nutrients-16-00911-f006:**
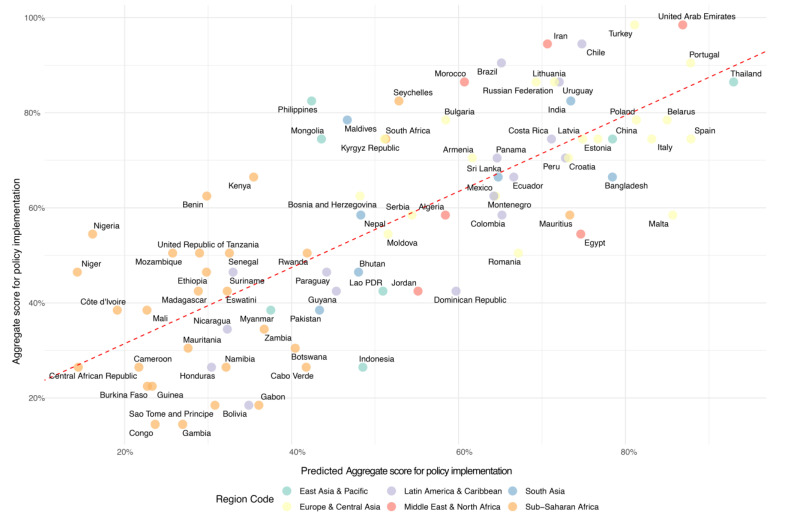
Predicted and actual aggregate scores for policy implementation in 2021 in 104 countries, differentiated by region (Note: The red dashed line represents the line of best fit in the linear regression).

**Table 1 nutrients-16-00911-t001:** Components of the nutrition aggregate score for policy implementation.

Section	Description
Health Information Systems	Functioning system for generating reliable cause-specific mortality data ^1^
Public Health Infrastructure	Existence of an Operational Unit, Branch, or Department in Ministry of Health with responsibility for NCDs ^2^
Partnerships and Multisectoral Collaboration	Existence of a national multisectoral commission, agency, or mechanism for NCDs ^2^
Strategy and Action Plans	Existence of a set of time-bound national targets based on WHO guidance ^1^
Strategy and Action Plans	Existence of an operational, multisectoral national NCD policy, strategy, or action plan ^1^
Nutrition Policy	Existence of operational policy/strategy/action plan for unhealthy diet ^2^
Nutrition Policy	Existence of tax on sugar-sweetened beverages ^2^
Nutrition Policy	Legislation/regulations fully implementing the International Code of Marketing of Breast Milk Substitutes ^1^
Nutrition Policy	Existence of any policies on marketing of foods to children ^1^
Nutrition Policy	Existence of national policies on saturated/ trans fatty acids ^1^
Nutrition Policy	Existence of any policies to reduce population salt consumption ^1^
Nutrition Policy	Existence of a tax on foods high in fat, sugars, or salt is implemented in the country. *
Nutrition Policy	Existence of price subsidies for healthy foods *
Public Education and Awareness	Implementation of a nutrition public awareness program ^2^
Monitoring and Surveillance	Conduct a STEPS survey or a comprehensive health examination survey every five years ^1^
Monitoring and Surveillance	A recent national adult risk factor survey is conducted to examine the prevalence of unhealthy diet *

^1^ World Health Organization (WHO) “Non-communicable Diseases Progress Monitor” for the years 2017, 2019, and 2021; ^2^ World Health Organization (WHO) The Global Health Observatory—Non-communicable Diseases: National Capacity” for the years 2017, 2019 and 2021. * Data only available for 2021. Indicators excluded from ASPI calculation to maintain comparability of countries and years.

**Table 2 nutrients-16-00911-t002:** Explanatory variables for policy implementation analysis.

Variable	Year	Description	Source	Coverage
Literacy Rate	2018 *	The percentage of people ages 15 and above who can both read and write, and understand a short simple statement about their everyday life. Used as a proxy for the effectiveness of the education system.	World Bank [20]	135 countries
Human Development Index (HDI)	2021	A summary measure of average achievement in key dimensions of human development: life expectancy, education, and standard of living.	UNDP [21]	188 countries
GDP per Capita (Thousands)	2021	Gross Domestic Product divided by midyear population, in current US dollars. Measures economic output per person, indicating economic prosperity and resource availability.	World Bank [22]	187 countries
Tax Burden	2021	Reflects marginal tax rates on income and the overall level of taxation as a percentage of GDP. A composite measure indicating the economic impact of taxation on individuals and corporations.	The Heritage Foundation [23]	175 countries
Current Health Expenditure per Capita (Thousands)	2020	Current expenditures on health per capita in current US dollars. Measures the economic resources allocated to health care per person.	World Bank [24]	185 countries
Universal Health Coverage (UHC) Index	2021	Coverage index for essential health services (0 to 100). Measures access to essential healthcare services—an aspect of Universal Health Coverage.	WHO [25]	194 countries
Premature Mortality from NCDs	2019	The percent of 30-year-old people who would die before their 70th birthday from cardiovascular disease, cancer, diabetes, or chronic respiratory disease. Indicates the burden of major NCDs on the population.	World Bank [26]	183 countries
Population ages 65 and Above	2022	The percentage of the total population that is 65 years and above. Indicates the demographic aging of the population and potential NCD burden.	World Bank [27]	192 countries
Total NCD Mortality Rate	2019	Age-standardized mortality rate from NCDs per 100,000 persons. Adjusts for differences in the age distribution of the population, indicating the overall NCD mortality burden.	WHO [28]	183 countries
Cost and Affordability of a Healthy Diet	2021	The cost of accessing the least expensive locally available foods to meet requirements for a healthy diet. Measures physical and economic access to healthy foods, crucial for preventing NCDs through nutrition.	FAO [29]	155 countries
Prevalence of Obesity among Adults	2016	Estimation of age-standardized percentage of adults with a BMI of 30 kg/m^2^ or higher. Indicates the prevalence of obesity, a major risk factor for several NCDs.	WHO [30]	190 countries
Prevalence of Hypertension among Adults	2019	Age-standardized prevalence of raised blood pressure among persons aged 18+ years, defined as systolic blood pressure ≥ 140 and/or diastolic blood pressure ≥90 mmHg. A major risk factor for coronary heart disease and stroke, indicating the burden of NCDs.	WHO [31]	192 countries
Total Alcohol Consumption per Capita	2019	Total amount of alcohol consumed per person (15 years of age or older) over a calendar year, in liters of pure alcohol. Associated with the risk of developing health problems such as NCDs.	World Bank [32]	187 countries
Added sugar intake per Capita	2018	Estimation of percent of total kcal per day (energy contribution). Represents the energy contribution from added sugars, highlighting diet-related NCD risk.	GDD [33]	183 countries
Sodium intake per Capita	2018	Estimation of intake per person in milligrams per day. Quantifies daily sodium intake, critical for understanding dietary risk factors for NCDs.	GDD [33]	183 countries
Saturated fat intake per Capita	2018	Estimation of percent of total kcal per day (energy contribution). Assesses energy contribution from saturated fats, which is important for evaluating diet-related NCD risk.	GDD [33]	183 countries

* for 49 countries, the last available data with a data break in 2015; BMI, Body Mass Index; GDD, Global Dietary Database; GDP, Gross Domestic Product; FAO, Food and Agriculture Organization of the United Nations; HDI, Human Development Index; NCD, non-communicable diseases; UHC, Universal Health Coverage; UNDP, United Nations Development Programme; WHO, World Health Organization.

**Table 3 nutrients-16-00911-t003:** National-level predictors for diet-related policy implementation: results of univariate and multivariate linear regression models.

	Univariate Linear Regression			Multivariate Linear Regression		
Variables of Interest	Effect Size (95% CI)	R^2^	*p*-Value	Effect Size (95% CI)	β	*p*-Value
Literacy Rate	0.600 (0.487, 0.713)	0.212	<0.001	−0.301 (−0.482, −0.120)	−0.239	0.001
HDI	99.249 (88.780, 109.719)	0.382	<0.001	14.381 (79.323, 201.438)	0.843	<0.001
GDP per Capita	0.232 (0.161, 0.303)	0.069	<0.001	−0.671 (−1.600, 0.258)	−0.260	0.156
Tax Burden	0.014 (−0.152, 0.180)	0.000	0.866	−0.183 (−0.428, 0.062)	−0.074	0.142
Health Expenditure per Capita	3.963 (3.020, 4.907)	0.110	<0.001	2.202 (−1.178, 14.582)	0.065	0.727
UHC Index	0.908 (0.812, 1.003)	0.375	<0.001	0.444 (0.087, 0.801)	0.299	0.015
Cost of a Healthy Diet	−4.949 (−7.575, −2.324)	0.029	<0.001	−4.358 (−7.507, −1.210)	−0.128	0.007
Inability to Afford Healthy Diet	−0.420 (−0.470, −0.371)	0.401	<0.001	−0.031 (−0.180, 0.118)	−0.046	0.682
Population > 65 years (%)	1.504 (1.246, 1.761)	0.186	<0.001	−0.093 (−0.819, 0.634)	−0.025	0.802
NCD Mortality Rate	0.047 (−0.057, −0.037)	0.124	<0.001	−0.083 (−0.156, −0.010)	−0.532	0.026
Premature NCD Mortality	−1.109 (−1.356, −0.863)	0.125	<0.001	2.348 (0.605, 4.092)	0.625	0.008
Hypertension Prevalence	−0.094 (−0.392, 0.205)	0.001	0.538	−0.250 (−0.637, 0.137)	−0.069	0.205
Obesity Prevalence	0.427 (0.256, 0.597)	0.041	<0.001	−0.194 (−0.545, 0.157)	−0.071	0.277
Alcohol Intake	1.739 (1.254, 2.223)	0.082	<0.001	−0.322 (−0.953, 0.308)	−0.052	0.315
Added Sugars Intake	0.319 (0.042, 0.596)	0.009	0.024	0.027 (−0.265, 0.319)	0.008	0.857
Sodium Intake	0.012 (0.010, 0.015)	0.119	<0.001	0.001 (−0.002, 0.004)	0.034	0.459
Saturated Fat Intake	0.927 (0.297, 1.556)	0.015	0.004	−0.390 (−1.117, 0.337)	−0.048	0.292

β, standardized coefficient; CI, confidence interval; GDP, Gross Domestic Product; HDI, Human Development Index; NCD, non-communicable diseases; R^2^, R-squared; UHC, Universal Health Coverage.

## Data Availability

The data presented in this study were derived from the following resources available in the public domain: [13,14,15,17,20,21,22,23,24,25,26,27,28,29,30,31,32,33].

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
