# Peer review of "Implementation of National Nutrition Policies and Strategies to Reduce Unhealthy Diets: An Ecological Analysis of 194 Countries from 2017 to 2021"

_nutrients, 2024, doi:10.3390/nu16060911_

Round 1

Reviewer 1 Report

Comments and Suggestions for Authors

The presented work, utilizing secondary data, outlines the implementation of selected dietary policies for the prevention of non-communicable chronic diseases in various countries worldwide. Its content is interesting, yet too lengthy for readers; it may be beneficial to consider presenting selected data for individual world regions according to the WHO division. Selected figures, due to the large amount of data, are not very readable, some have not been correctly described, e.g., Figure 1 (lack of legend). In the description of the results, the figure number to which they refer is not mentioned (line 280). The conclusions are rather another summary of the results and require shortening.

Author Response

We extend our sincerest gratitude to the reviewer for their thorough evaluation of our manuscript titled "Implementation of National Nutrition Policies and Strategies to Reduce Unhealthy Diets: A Geopolitical Analysis of 194 Countries". Your insightful comments have significantly contributed to the refinement of our research paper. Below, we provide detailed responses to each of the points raised in your review, along with explanations of the adjustments made to our manuscript. These amendments have been highlighted in the revised document for ease of identification.

2. Point-by-point Response to Comments and Suggestions for Authors

Comment 1: The reviewer noted that while the work is interesting, the content is excessively lengthy and suggested considering the presentation of selected data for individual world regions according to the WHO division.
Response 1: We acknowledge the reviewer's concern regarding the lengthiness of our content and the suggestion to present data according to WHO world regions. In response, we have maintained the use of data from all WHO Member States, believing this comprehensive approach adds substantial value to our analysis. However, to address the concern, we have presented the data from the World Bank Income Group and WHO Region, with supplementary materials covering detailed analyses of all countries. This approach allows for a more focused yet comprehensive overview of the geopolitical variations in nutrition policy implementation without sacrificing the depth or breadth of our findings. 

Comment 2: The reviewer pointed out issues with the manuscript’s readability and descriptions of figures, specifically citing a lack of legend for Figure 1.
Response 2: We appreciate the reviewer highlighting the technical issues with the figures in our initial submission. To rectify this, we have replaced all figures with high-quality PNG versions, ensuring that details are preserved and clearly visible in the manuscript. Specifically, for Figure 1, we have now included a comprehensive legend that accurately describes its components. This amendment enhances the clarity and interpretability of our visual data, facilitating a better understanding of our research outcomes.

Comment 3: The figure numbers to which they refer were not mentioned in the results’ description.
Response 3: Thank you for pointing out this oversight. We have revised the results section to explicitly mention all references to figures, providing a direct correlation between the textual and visual data. This change aids readers in navigating our findings with greater ease and enhances the manuscript's coherence. These corrections can be found starting on page 10, line 498.

Comment 4: The reviewer critiqued that the conclusions were more of a summary of the results than succinct conclusions, recommending shortening.
Response 4: We concur with the reviewer's assessment regarding the original conclusions drawn from our study. In light of this, we have significantly condensed the conclusions section to encapsulate the core findings and implications of our research succinctly. The revised conclusions now more directly reflect the positive trends observed in the implementation of national nutrition policies against diet-related NCDs while also acknowledging the persistent challenges and disparities. This revision clarifies the necessity for a context-specific, multidisciplinary approach to policy development, emphasizing the importance of integrated, evidence-based strategies. The streamlined conclusions can be found on page 15, in the concluding paragraph.

Reviewer 2 Report

Comments and Suggestions for Authors

This is an important and timely analysis of worldwide implementation of and improvement of national nutritional policies toward NCDs.  The English is perfect.  The analyses and conclusions are sound.  A few suggestions are given concerning naming variables and concerning figures.

Abstract

Line 16: The authors state, “The mean score in 2021 was 51.98% (SD = 23.94)”.  The two digits to the right of the decimal point convey a level of precision that is not in the raw data.  With a standard deviation of 24, the mean could just as well be written as 52 without a loss of actual information.  This is a common problem in scientific manuscripts and is common in this paper.  I suggest that the authors examine the statistics they present, especially the percentages, and consider whether the extra decimals are necessary.  This can easily be accommodated since none of these values are from figures or charts and are only part of the text.

Methods

Line 106: “WHO” has already been defined as an abbreviation for World Health Organization in line 42. “NCD” has already been defined as an abbreviation for non-communicable diseases in line 32 and “CCS” has already been defined as an abbreviation for Country Capacity Surveys in line 64.

Lines 124-136: The Aggregate Score as defined here is one of the major advances of this manuscript.  It is not validated in the usual sense that new published metrics are verified.  While the data environment is certainly unique, it would be helpful, probably in the Discussion, to elaborate some on the logic of this score and to explain the process of its development and to compare it with similar scores.

Table 1: There are 13 sections to the Aggregate Score that are identified here.  Would it be possible to give some indication of what fraction of the score came from each of these sections?  The composition of this score is still fairly opaque.

Table 2 is very helpful in understanding the source and types of data used in the explanatory models.  One variable that was highlighted in the text was the number of missing values from the sources.  Would it be possible to indicate in the table the number of missing values or number of included values for each source?

Results

Lines 201-206: Again, a consideration of the number of digits needed to the right of the decimal point.  Do we really need 3 digits for the SD when the SD is half of the mean?  While the ANOVA is understandably not significant, would a paired t-test have been more appropriate here?

Lines 205-206: The authors state, “the lowest scores marginally increasing to 3.85%”.  This is a bit confusing.  As a casual reader, I take this to mean that among those countries with the lowest quartile of aggregate scores in 2019, they increased their scores by an average of 3.85% in 2021.  Is that the correct interpretation?  A little more detail would be helpful.

Figure 1: This may be a problem with the provisional manuscript but there is no indication in the figure what the 16 bars represent.  What the colors represent is also missing.  It is not enough to point to Table S1, which is not available to the casual reader.

Lines 229-244 and Figure 2: This is rather confusing.  Is the “diet policy score” the same as the Aggregate Score or the “implementation score” or the “Diet Policy Index”?  A word search shows that the phrase “diet policy score” is used only once in the text on line 243.  “Diet Policy Index” is used in Figure 2 for the X-axis.  But if they are the same, then the values are confusing, since the mean Aggregate Score is 49 with SD= 24 and the X-axis scale on the graph runs only to 12.5.  However, in the text, Turkey, Finland, and UAR are said to have implementation scores of 96.15%, not 12.5.  However, when Ukraine is described, its 2021 implementation score is said to be 10, which corresponds to the figure.

This raises a more general issue that should be addressed.  This new Aggregate Score is called different things at different places in the text.  I believe that it is called the “implementation score” here and elsewhere.  As consistent name would be helpful.

The authors will notice that I am ignoring Supplementary tables.  A manuscript should be self-contained and definable on its own term.

Lines 253-257: Again, with N=194 and variable is 0 or 1, two digits to the right of the decimal place give a false sense of the accuracy of these values.

Figures 1 and 3.  Both of these figures show Nutrition-Related Policies and Strategies.  However, Figure 1 has 16 bars and Figure 3 has 13 groups of bars.  Why the difference?

Figure 4 has the same problem.  What is the Diet Policy Score?  The values of 3.81 – 8.43 do not correspond to the Aggregate Score.  They also do not correspond well to increases in the “Diet Policy Index” from Figure 2 where the highest scores were only about 7.5, not 8.43.  I also suspect that the Caption is not right.  “Mean implementation scores for each of the Nutrition-Related Policies and Strategies by World Bank Income Group in 2021.”  This suggests multiple values for each of the Income Groups.

Figure 5: Same issues.

Lines 278-279: The authors state, “Aggregate scores progressively increased from low-income (mean = 3.62, SD = 1.99) to high-income countries (mean = 8.36, SD = 2.55)”.  However, in the Abstract the authors state, “The mean [implementation] score in 2021 was 51.98% (SD = 23.94)”.  How can the overall mean be 52 if the low countries average 4 and the high countries average 8?

Line 292: HDI not defined.

Line 295: UHC not defined.

Table 3: If GDP per Capita were expressed in larger units, these values of “0.000 (0.000, 0.000)” would appear more sensible.  The same for Current Health Expenditure per Capita and Sodium Intake.  The additional variables in the multivariate analyses were not described in the Methods or as a footnote to this table.  Please correct.

Figure 6: The Region Codes do not clearly show up in the figure.

Data Availability Statement

It would be helpful if these was numbered in the same fashion as the Reference list.  It would be easier to read and to use.

References

Reference 1: The suggested author is “GBD 2019 Diseases and Injuries Collaborators.”

Reference 2 has no link or URL.

Reference 9 has no link or URL.

Author Response

For the research article "nutrients-2924460."

Response to Reviewer 2 Comments

1. Summary

Thank you very much for your insightful comments and suggestions regarding our manuscript titled "Implementation of National Nutrition Policies and Strategies to Reduce Unhealthy Diets: A Geopolitical Analysis of 194 Countries." We appreciate the time and effort you have devoted to reviewing our work. Your feedback has been invaluable in enhancing the quality and clarity of our research. Please find below detailed responses to each of the points raised in your review, along with corresponding revisions highlighted in track changes in the re-submitted files. We have made every effort to address your concerns comprehensively and have provided explanations for the changes made to the manuscript.

2. Point-by-point Response to Comments and Suggestions for Authors

Comment 1: Abstract: Line 16: The authors state, “The mean score in 2021 was 51.98% (SD = 23.94)”. The two digits to the right of the decimal point convey a level of precision that is not in the raw data. With a standard deviation of 24, the mean could just as well be written as 52 without a loss of actual information. This is a common problem in scientific manuscripts and is common in this paper. I suggest that the authors examine the statistics they present, especially the percentages, and consider whether the extra decimals are necessary. This can easily be accommodated since none of these values are from figures or charts and are only part of the text.

Response 1: Thank you for your feedback regarding the statistical reporting in our manuscript. After reviewing your comments, we agree that the extra decimal places do not help readers understand our findings better. Instead, they might suggest a level of accuracy that is not supported by the data's nature and standard deviation. We have revised the statistics throughout the manuscript to reflect an appropriate level of precision. For example, in the abstract, we have adjusted the mean score for 2021 from "51.98% (SD = 23.94)" to "52% (SD = 24)," as you suggested. We have made similar changes consistently throughout the manuscript to ensure that the data presentation is clear and precise, without implying unwarranted accuracy.

Comment 2: Line 106: “WHO” has already been defined as an abbreviation for World Health Organization in line 42. “NCD” has already been defined as an abbreviation for non-communicable diseases in line 32 and “CCS” has already been defined as an abbreviation for Country Capacity Surveys in line 64.

Response 2: Our manuscript has been updated to improve readability for our readers. We have made a specific change to the text on line 106. Instead of saying "by utilizing data from the World Health Organization's (WHO) Non-communicable Diseases (NCD) Progress Monitors and Country Capacity Surveys (CCS)," we now say "By using data from the WHO's NCD Progress Monitors and CCS, we gain insights into national efforts and their outcomes in addressing dietary health challenges." This change adheres to best practices in academic writing by using abbreviations consistently after their initial definition.

Comment 2: Line 106: “WHO” has already been defined as an abbreviation for World Health Organization in line 42. “NCD” has already been defined as an abbreviation for non-communicable diseases in line 32 and “CCS” has already been defined as an abbreviation for Country Capacity Surveys in line 64.

Response 2: We are grateful for your attention to detail in identifying the redundant abbreviations in our manuscript. To improve readability and consistency, we have made revisions to the text. We now use the abbreviations of WHO, NCD, and CCS consistently throughout the document. This change eliminates repetition and makes the narrative easier to read. Specifically, on line 106, we have updated the text from "by utilizing data from the World Health Organization's (WHO) Non-communicable Diseases (NCD) Progress Monitors and Country Capacity Surveys (CCS)" to "By using data from the WHO's NCD Progress Monitors and CCS, we gain insights into national efforts and outcomes in addressing dietary health challenges."

Comment 3: Lines 124-136: The Aggregate Score as defined here is one of the major advances of this manuscript. It is not validated in the usual sense that new published metrics are verified. While the data environment is certainly unique, it would be helpful, probably in the Discussion, to elaborate some on the logic of this score and to explain the process of its development and to compare it with similar scores.

Response 3: We have elaborated on the Aggregate Score (ASPI) in the Discussion section of our manuscript, in response to your insightful suggestion. The newly added text explains the ASPI's methodological foundation, distinction from existing frameworks, and specific focus on evaluating the implementation of dietary strategies. By drawing comparisons with existing metrics, we clarify the unique contribution of our methodology to the field of nutrition policy research. A detailed explanation of the ASPI's development and rationale for its specific focus can now be found in the revised manuscript.

Comment 4: Table 1: There are 13 sections to the Aggregate Score that are identified here. Would it be possible to give some indication of what fraction of the score came from each of these sections? The composition of this score is still fairly opaque.

Response 4: We have updated Table 1 to provide a clearer view of how each of the 13 sections contributes to the overall Aggregate Score for Policy Implementation (ASPI). This update includes a detailed breakdown of the score's components, indicating the weight each section holds in the calculation of the ASPI. We have also explained the methodology behind the ASPI calculation in the revised manuscript. To aid in understanding, we have normalized the scores against a theoretical maximum and converted the results into percentages for comparative analysis. This approach ensures transparency in how the ASPI reflects the implementation scores of all diet-related policies across 13 indicators.

Comment 5: Table 2 is very helpful in understanding the source and types of data used in the explanatory models. One variable that was highlighted in the text was the number of missing values from the sources. Would it be possible to indicate in the table the number of missing values or number of included values for each source?

Response 5: We appreciate your positive feedback on Table 2 and your suggestion to improve it by including data on missing values. We have updated Table 2 to add a new column called "Coverage" which shows the number of countries for each variable. This change will help you see how complete the dataset is and the extent of missing information for each source. To make our analysis easier to read, we have removed any specific mentions of missing data from the text. The updated table now provides enough information about missing data. This change ensures that the manuscript focuses on the analysis while still offering detailed insight into the dataset's robustness and coverage.

Comment 6: Lines 201-206: Again, a consideration of the number of digits needed to the right of the decimal point. Do we really need 3 digits for the SD when the SD is half of the mean? While the ANOVA is understandably not significant, would a paired t-test have been more appropriate here?

Response 6: Thanks for your feedback on our manuscript. We have adjusted the SD values to two decimal places and used ANOVA to compare mean scores across multiple years, World Bank Income Groups, and WHO geographical regions. We've also done a t-test to compare the means between 2017 and 2021, and found no significant differences.

Comment 7: The authors state, “the lowest scores marginally increasing to 3.85%”. This is a bit confusing. As a casual reader, I take this to mean that among those countries with the lowest quartile of aggregate scores in 2019, they increased their scores by an average of 3.85% in 2021. Is that the correct interpretation? A little more detail would be helpful.

Response 7: Thank you for highlighting the need for clarity regarding the interpretation of the scores. We acknowledge that the original phrasing could lead to confusion about the improvements in the aggregate scores among the lower-quartile countries from 2019 to 2021. To address this, we have revised the wording to provide a clearer explanation of the score changes observed. The revised text now states: "The data for 2021 reveals significant variation in the adoption of dietary policies aimed at reducing non-communicable diseases across the 194 countries examined. Notably, the United Arab Emirates, Turkey, and Finland each achieved an implementation score of 96.15% (12.5 points out of a total of 13). Conversely, countries at the lower end of the spectrum, such as Yemen, Somalia, and Libya, recorded implementation scores as low as 3.85% (0.5 points)." This adjustment aims to articulate more precisely the distribution of scores in 2021, enhancing the reader's understanding of the range and implications of the data presented.

Comment 8: This may be a problem with the provisional manuscript but there is no indication in the figure what the 16 bars represent. What the colors represent is also missing. It is not enough to point to Table S1, which is not available to the casual reader.

Response 8: We appreciate you for pointing out the issues in Figure 1's presentation. Multiple figures lacked clarity due to a tachnical error during the export process from word to pdf. We understand the importance of making our figures self-explanatory and accessible to all readers. Therefore, we revised the figure by incorporating clear labels and legends within Figure 1. Additionally, we have addressed the issue of lost details during the export process by converting all figures from PDFs to high-quality PNG formats. This ensures that all figures are presented with optimal clarity and precision in the final manuscript.

Comment 9: Lines 229-244 and Figure 2: This is rather confusing. Is the “diet policy score” the same as the Aggregate Score or the “implementation score” or the “Diet Policy Index”? A word search shows that the phrase “diet policy score” is used only once in the text on line 243. “Diet Policy Index” is used in Figure 2 for the X-axis. But if they are the same, then the values are confusing, since the mean Aggregate Score is 49 with SD= 24 and the X-axis scale on the graph runs only to 12.5. However, in the text, Turkey, Finland, and UAR are said to have implementation scores of 96.15%, not 12.5. However, when Ukraine is described, its 2021 implementation score is said to be 10, which corresponds to the figure. This raises a more general issue that should be addressed. This new Aggregate Score is called different things at different places in the text. I believe that it is called the “implementation score” here and elsewhere. A consistent name would be helpful. The authors will notice that I am ignoring Supplementary tables. A manuscript should be self-contained and definable on its own terms.

Response 9: We apologize for any confusion caused by the inconsistent terminology used to describe the scoring system in our research. To address this issue, we have reviewed the manuscript and made changes to ensure consistent use of terminology when referring to our scoring system. We have adopted the term "Aggregate Score for Policy Implementation (ASPI)" throughout the text, including in the descriptions of figures where relevant. This refers to the refined score, which we present as a percentage of the theoretical maximum of 13 points to provide a more intuitive understanding of our findings. The ASPI is therefore the normalized score converted into percentages against the theoretical maximum of 13 points. We have made this clear in the manuscript to avoid any confusion about the score's calculation and representation. We have also revised the figures and related discussions in the text to ensure that all references to the scoring system are consistent and accurately reflect the methodology of our analysis.

Comment 10: Lines 253-257: Again, with N=194 and variable is 0 or 1, two digits to the right of the decimal place give a false sense of the accuracy of these values.

Response 10: We appreciate the opportunity to provide further clarification on our methodology for calculating the mean changes for various health policies, as reported in our manuscript. This calculation considers both positive (+1) and negative changes (-1_, with the denominator being the total number of countries for which data was available, thereby accurately reflecting the global landscape of policy evolution.

Here's the revised explanation of our calculation process:

  • Positive Changes: The number of countries that have strengthened or introduced new policies.
  • Negative Changes: The number of countries that have weakened or removed existing policies.
  • Total Countries with Available Data: This is the denominator in our calculation, ensuring that the percentage change reflects the proportion of all countries assessed, not just those with policy changes.

The calculation method thus accounts for the full spectrum of policy evolution, acknowledging both advancements and regressions in public health policy across the countries. This approach allows us to present a balanced and comprehensive picture of global policy trends, emphasizing the overall direction of policy changes in relation to unhealthy diets and NCD prevention. Our calculation for the mean percentage change for policies such as the SSB tax, multisectorial commissions, and NCD operational units, thereby, provides a clear indication of how widespread the adoption, reinforcement, or rollback of these policies has been among the countries evaluated. This methodological clarity ensures that our analysis accurately captures and communicates the complexity of global health policy landscapes, reflecting our commitment to rigorous and transparent research practices.

Comment 11: Figures 1 and 3 both show Nutrition-Related Policies and Strategies. However, Figure 1 has 16 bars and Figure 3 has 13 groups of bars. Why the difference?

Response 11: Thank you for highlighting the discrepancy between Figures 1 and 3 in our manuscript. The difference in the number of bars across these figures is attributable to the availability of data for specific nutrition-related policies and strategies. In the revised text and Table 1, we clarify that data for three policies were available only for the year 2021:

  1. The existence of a tax on foods high in fat, sugars, or salt.
  2. The existence of price subsidies for healthy foods.
  3. The conduct of a recent national adult risk factor survey to examine the prevalence of unhealthy diets.

These indicators were not included in the calculation of the Aggregate Score for Policy Implementation (ASPI) to ensure comparability across countries and years. Instead, they were incorporated solely for descriptive analysis concerning the policy implementation status in 2021. This methodological decision was made to maintain the integrity of our longitudinal analysis while also providing a snapshot of the latest developments in national nutrition policies. By doing so, we aimed to balance the need for a comprehensive assessment of policy implementation over time with the inclusion of emerging policy measures in our analysis.

Comment 12: Figure 4 has the same problem. What is the Diet Policy Score? The values of 3.81 – 8.43 do not correspond to the Aggregate Score. They also do not correspond well to increases in the “Diet Policy Index” from Figure 2 where the highest scores were only about 7.5, not 8.43. I also suspect that the Caption is not right. “Mean implementation scores for each of the Nutrition-Related Policies and Strategies by World Bank Income Group in 2021.” This suggests multiple values for each of the Income Groups. Figure 5: Same issues.

Response 12: We appreciate your detailed observation concerning Figure 4 and the need for clarification on the Diet Policy Score and its correlation with the Aggregate Score for Policy Implementation (ASPI). Upon reviewing your comments, we recognized that the discrepancy arises from the dual scoring systems employed in our analysis: the raw score and the normalized score. To address this, we have undertaken a thorough revision of the manuscript and figures to ensure consistency and clarity. The term "Diet Policy Score" was used inconsistently and has now been standardized to "Aggregate Score for Policy Implementation (ASPI)" throughout the document. This score reflects the cumulative impact of national nutrition policies and strategies, calculated as follows:

  • Raw Score: A simple sum of the implementation scores for all thirteen diet-related policies, creating an overall aggregate policy implementation score.
  • Normalized Score: This raw score is then converted to percentages against the theoretical maximum of 13 points, providing a normalized score that facilitates comparison across countries and over time.

The discrepancy in values observed in Figure 4, and the inconsistency with Figure 2, resulted from the presentation of raw scores. In response, we have revised both the text and figures to exclusively feature the normalized score in percentages. This approach enhances the manuscript's coherence and ensures that the figures accurately represent the data discussed in the text. Furthermore, the mean aggregate scores presented in Figures 4 and 5 reflect averages for specific groups, such as the World Bank Income Regions and WHO Geographical regions, and not the mean country average aggregate scores. We have clarified this distinction in the revised manuscript to prevent any confusion.

Comment 13: Lines 278-279: The authors state, “Aggregate scores progressively increased from low-income (mean = 3.62, SD = 1.99) to high-income countries (mean = 8.36, SD = 2.55)”. However, in the Abstract the authors state, “The mean [implementation] score in 2021 was 51.98% (SD = 23.94)”. How can the overall mean be 52 if the low countries average 4 and the high countries average 8?

Revised Response with Emphasis on Normalization:

Acknowledging your insightful query, we realized the necessity to clarify the transition from raw scores to normalized scores in our manuscript. Initially, the disparity in scores, as highlighted in your comment, stemmed from the presentation of raw scores, which did not provide a directly comparable measure across countries of differing economic statuses or regional backgrounds.

In response to this, we have meticulously revised our analysis and the manuscript to emphasize the normalization of scores. Specifically, we transitioned from presenting raw scores to utilizing normalized scores, calculated as percentages of the maximum possible score. This shift to normalized scores enables a uniform comparison of policy implementation effectiveness across all countries, irrespective of their income level or geographical region.

The revised manuscript now clearly states that the ASPI progressively increased from low-income countries (mean = 29.3%, SD = 14.9) to high-income countries (mean = 64.8%, SD = 20.5), reflecting a significant association between a nation's economic capacity and its implementation of nutrition policies. This normalization process addresses the earlier confusion by providing a coherent basis for comparison, evident in the comprehensive range of ASPI scores presented.

Moreover, our updated regional analysis further benefits from this normalization, revealing significant disparities in ASPI among geopolitical regions, with results highlighting the effectiveness of nutrition policies from the lowest in Sub-Saharan Africa (mean = 32.93%, SD = 18.26) to the highest in North America (mean = 75.0%, SD = 2.71). These normalized scores, supported by ANOVA results, offer a refined insight into the impact of economic and regional factors on policy implementation, aligning with our objective to present a clear, comparative analysis across the global landscape.

Comment 14: Line 292: HDI not defined; Line 295: UHC not defined.

Response 14: We acknowledge the oversight in not defining the Human Development Index (HDI) and Universal Health Coverage (UHC) upon their first mention in the manuscript. Recognizing the importance of clarity and comprehensiveness in academic writing, especially in a field as intricate as public health, we have revised the manuscript accordingly.

Comment 15: Table 3: If GDP per Capita were expressed in larger units, these values of “0.000 (0.000, 0.000)” would appear more sensible. The same for Current Health Expenditure per Capita and Sodium Intake. The additional variables in the multivariate analyses were not described in the Methods or as a footnote to this table. Please correct.

Response 15: We are grateful for your observation regarding the presentation of GDP per Capita, Current Health Expenditure per Capita, and Sodium Intake values in Table 3. Your suggestion to express these figures in larger units for improved clarity and readability was well-received. Accordingly, we have revised the representation of these economic indicators. GDP per Capita and Current Health Expenditure per Capita are now expressed in thousands of US dollars (USD), enhancing their interpretability.

Furthermore, we have described all variables used in the multivariate analyses within the Methods section and in Table 2 (Explanatory Variables for Policy Implementation Analysis). This includes a detailed account of how each variable was measured and sourced and its relevance to the study’s objectives.

Comment 16: Figure 6: The Region Codes do not clearly show up in the figure.

Response 16: We appreciate your patience and understanding regarding the technical issue that affected the visibility of region codes in Figure 6. The problem stemmed from the export process of the manuscript, wherein details in the figures, particularly the region codes, were compromised due to the use of PDF formats in the Word document template.

In response to this issue, we have taken corrective action by replacing all figures initially inserted as PDFs with high-quality PNG images. This adjustment ensures that the figures retain their clarity and detail throughout the manuscript export process, effectively resolving the visibility issue of the region codes in Figure 6.

Comment 17: Data Availability Statement: It would be helpful if these was numbered in the same fashion as the Reference list. It would be easier to read and to use.

Response 17: We appreciate your suggestion to enhance the readability and usability of the Data Availability Statement by adopting a numbered format akin to the Reference list. In line with your recommendation, we have revised the Data Availability Statement to present the information in a numbered list format.

Comment 18: Reference 1: The suggested author is “GBD 2019 Diseases and Injuries Collaborators.”; Reference 2 has no link or URL; Reference 9 has no link or URL.

Response 18: We thank you for pointing out the need for accuracy and completeness in our references. Following your feedback, we have reviewed and updated our references to ensure their correctness and to provide complete information.

Reviewer 3 Report

Comments and Suggestions for Authors

This is a well designed study and a very well written manuscript.  The study clearly is important and will add to the literature.  I just have a few suggestions:

1.      Methods:  the authors select a scoring system of 1. 0.5, o.  While this seems very reasonable, the authors should offer a rationale why this system was selected.  Asr there other examples in the literature?

2.      Thirteen components were selected from the QHO survey.  Were these all the components of the WHO survey or were just some selected and if so why?

3.      In figure 2., are all 194 countries represented?   That might be mentioned in the legend.

4.      I could not find Figure 4 mentioned in the Results Section

5.      If available, it would be nice to see if disease burden either in the form of obesity, type two diabetes or cardiovascular disease was a predictor of implementation.  If not, some mention and speculation would be appropriate

Author Response

1. Summary        
Thank you for the opportunity to address the comments from the peer review. Our responses to each point are provided below, reflecting the revisions made to our manuscript titled "Implementation of National Nutrition Policies and Strategies to Reduce Unhealthy Diets: A Geopolitical Analysis of 194 Countries."

2. Point-by-point Response to Comments and Suggestions for Authors
Comment 1: Methods: The authors select a scoring system of 1, 0.5, 0. While this seems very reasonable, the authors should offer a rationale why this system was selected. Are there other examples in the literature? Thirteen components were selected from the WHO survey. Were these all the components of the WHO survey, or were just some selected, and if so, why?
Response 1: In response to the reviewer's inquiry regarding our scoring system and the selection of the thirteen components from the WHO survey, we have made comprehensive updates to clarify our methodology. 
•    Table 1: Now includes a detailed breakdown of how each of the 13 sections contributes to the overall Aggregate Score for Policy Implementation (ASPI). 
•    Methodology Section: Enhanced with a thorough explanation of the ASPI calculation and the selection criteria for WHO survey components, providing clarity on the study's analytical framework.
•    Supplementary Materials: All indicators from the WHO’s NCD Progress reports and Country Capacity Surveys (CCS) have been listed, with an explanation on the inclusion and exclusion criteria applied, ensuring comprehensive transparency in our methodological approach.
•    Discussion section: we have added a paragraph mentioning: “In this study, we introduce the methodology for an aggregate score (ASPI) to evaluate the implementation of national nutrition policies aimed explicitly at mitigating the impact of unhealthy diets. This endeavor is distinct from previous frameworks, notably diverging from the WHO's Nutrition Governance Score [37], which, while comprehensive, has not been updated since 2018. Furthermore, it refines the scope beyond the broader lens of NCD policy evaluation found in Allen et al.'s work [18,19] by focusing exclusively on dietary strategies. Utilizing WHO's robust data, ASPI offers a focused, evidence-backed insight into the effectiveness of diet-related policy implementation, filling a crucial gap in the literature. Through this refined lens, our study contributes to the critical discourse on enhancing nutrition policy frameworks.” 
•    37. World Health Organization (WHO). Nutrition Landscape Information System (NLiS): Nutrition Governance Score. Available online: https://www.who.int/data/nutrition/nlis/info/nutrition-governance-score (accessed on March 16). 
•    18. Allen, L.N.; Nicholson, B.D.; Yeung, B.Y.T.; Goiana-da-Silva, F. Implementation of non-communicable disease policies: a geopolitical analysis of 151 countries. Lancet Glob Health 2020, 8, e50-e58, doi:10.1016/S2214-109X(19)30446-2. 
•    19. Allen, L.N.; Wigley, S.; Holmer, H. Implementation of non-communicable disease policies from 2015 to 2020: a geopolitical analysis of 194 countries. Lancet Glob Health 2021, 9, e1528-e1538, doi:10.1016/S2214-109X(21)00359-4.

Comment 2: In Figure 2, are all 194 countries represented? The legend might mention that.
Response 2: The title of Figure 2 has been updated to "The Global Aggregate Score for Policy Implementation in 2021 in 194 countries, differentiated by World Bank Income Group, mapped against the percentage change in the score since 2017." This revision explicitly indicates that all 194 countries are represented in the analysis, providing a comprehensive global overview of policy implementation scores across different income groups. 

Comment 3: I could not find Figure 4 mentioned in the Results Section.
Response 3: We acknowledge the oversight regarding the mention of Figure 4 in the Results Section and have corrected this in the revised manuscript. Additionally, to address the issue of lost details during the export process, all figures, including Figure 4, have been converted from PDFs to high-quality PNG formats. This ensures that each figure is presented with optimal clarity and precision, facilitating a better understanding of the data and findings conveyed in our study.

Comment 4: If available, it would be nice to see if disease burden either in the form of obesity, type two diabetes, or cardiovascular disease was a predictor of implementation. If not, some mention and speculation would be appropriate.
Response 4: In the revised manuscript, we have included an analysis of two separate Non-Communicable Disease (NCD)-related variables: obesity prevalence and hypertension prevalence. These variables were chosen for their well-documented role as risk factors contributing to the overall NCD burden. The inclusion of these specific indicators allows for a nuanced exploration of the relationship between disease burden and the implementation of national nutrition policies. While other NCD burden variables were initially considered, they were ultimately excluded due to significant overlap with the chosen indicators, such as NCD mortality rate and premature NCD mortality rate. This decision was made to avoid redundancy and ensure the clarity and precision of our analysis. 
